# Microstructural Transformation and Hydrogen Generation Performance of Magnesium Scrap Ball Milled with Devarda’s Alloy

**DOI:** 10.3390/ma15228058

**Published:** 2022-11-15

**Authors:** Olesya A. Buryakovskaya, Mikhail S. Vlaskin

**Affiliations:** Laboratory of Energy Storage Substances, Joint Institute for High Temperatures of the Russian Academy of Sciences, 125412 Moscow, Russia

**Keywords:** magnesium scrap, Devarda’s alloy, ball milling, microstructure, hydrogen production

## Abstract

A method for magnesium scrap transformation into highly efficient hydroreactive material was elaborated. Tested samples were manufactured of magnesium scrap with no additives, or 5 and 10 wt.% Devarda’s alloy, by ball milling for 0.5, 1, 2, and 4 h. Their microstructural evolution and reaction kinetics in 3.5 wt.% NaCl solution were investigated. For the samples with additives and of scrap only, microstructural evolution included the formation of large plane-shaped pieces (0.5 and 1 h) with their further transformation into small compacted solid-shaped objects (2 and 4 h), along with accumulation of crystal lattice imperfections favoring pitting corrosion, and magnesium oxidation with residual oxygen under prolonged (4 h) ball milling, resulting in the lowest reactions rates. Modification with Devarda’s alloy accelerated microstructural evolution (during 0.5–1 h) and the creation of ‘microgalvanic cells’, enhancing magnesium galvanic corrosion with hydrogen evolution. The 1 h milled samples, with 5 wt.% Devarda’s alloy and without additives, provided the highest hydrogen yields of (95.36 ± 0.38)% and (91.12 ± 1.19)%; maximum reaction rates achieved 470.9 and 143.4 mL/g/min, respectively. Such high results were explained by the combination of the largest specific surface areas, accumulated lattice imperfections, and ‘microgalvanic cells’ (from additive). The optimal values were 1 h of milling and 5 wt.% of additive.

## 1. Introduction

At present, there is no doubt that the transition from fossil fuels toward renewable energy sources is not merely an issue of ecological safety, but a matter of sustainable energy security and development as well. The current renewable energy share in the global energy consumption is relatively small: about 10% only. Nevertheless, it is forecasted to rise to 17–28% by 2030, and contribute as much as 25–68% by 2050 [1]. To reach these targets, a problem of efficient renewable energy storage and distribution should be solved—and hydrogen is considered as an ultimate solution. Hydrogen represents a promising green energy carrier that can be obtained using various energy sources: solar, wind, geothermal or nuclear energy, hydroelectric and ocean thermal energy conversion power, etc. [2]. It can be implemented in the buildings, transportation, and industry sectors [3]. Its energy (lower heating value—120 MJ/kg) can be effectively converted into electricity by means of internal combustion engines, hydrogen-fired gas turbines, and polymer electrolyte membrane (PEM) fuel cells [4,5,6].

Although the benefits from the use of hydrogen fuel are amazing, a number of significant challenges exist so far. One of the main problems is leakage and diffusion which can lead to the formation of hazardous hydrogen–air mixtures, which are flammable and explosive at hydrogen concentrations of 4–75% and 18–59% respectively [7]. Another considerable drawback is hydrogen embrittlement, resulting in construction steel degradation that makes hydrogen troublesome for pipe transportation [8]. The most conventional hydrogen storage technologies include physical storage in a compressed or liquid form and chemical storage (generally in form of metal hydrides). Their disadvantages include correspondingly tremendous pressures (up to 700 or 1000 bar), extremely low temperatures (down to 20 K), and relatively low hydrogen storage densities, slow kinetics, low reversibility, and high dehydrogenation temperatures [9].

An alternative approach to hydrogen storage, transportation and delivery is the implementation of hydroreactive metals. With maximum hydrogen yields of 111 and 83 g (per kg) respectively, aluminum and magnesium represent almost perfect intermediate energy carriers. The option of using aluminum- or magnesium-based waste materials (beverage cans, foils, wires, white dross, and other scrap [10,11,12,13,14,15,16,17,18,19]) makes the said approach even more attractive. However, at normal conditions, aluminum is protected from oxidation by a dense passive oxide film, and magnesium’s reaction with pure water is rapidly interrupted due to the formation of a low-permeable product, Mg(OH)_2_. Therefore, to induce hydrogen generation, some chemical or physical activation methods should be applied [20,21,22,23,24]. The most extensively investigated ones include high reaction temperatures (above 100 °C) [25,26,27,28,29,30,31,32], various acidic [33,34,35], alkali (for aluminum) [36,37,38,39] and salt (NaCl, KCl, MgCl_2_, NiCl_2_, CoCl_2_, CuCl_2_, AlCl_3_) [40,41,42,43,44,45,46,47,48] solutions, liquid metal embrittlement (for aluminum) [49,50,51], alloying with metals like Ca, Ni, Sn, Fe, Li, Zn, Bi, Cu (brand-new or recovered from electronic and electrical waste) [52,53,54,55,56,57,58,59,60,61], preparation of composite powders with metal (Ni, Nd, Bi, Zn, In, Wood’s alloy, etc.) [62,63,64,65,66], and non-metal (e.g., NaCl, AlCl_3_, Bi_2_O_2_CO_3_, Bi(OH)_3_, Al(OH)_3_, Al_2_O_3_) [67,68,69] additives, or their mixtures (e.g., carbon materials with Bi, Ni, Cu, Co, MgCl_2_, AlCl_3_, Ga-based eutectic alloys, Ga–In–Sn alloy with NiCl_2_ and CoCl_2_, etc.) [70,71,72,73,74,75,76,77,78], ball milling without additives [79,80,81], and particle size reduction [82,83,84].

Aluminum and magnesium are widely used for structural applications due to their high specific strength, low density, good thermal conductivity and castability. About 38% of the total magnesium alloys production is consumed by the manufacture of die cast parts and wrought products (sheets and profiles), and the machining process of castings and sheets results in a large amount of magnesium waste chips and discards with nearly 30% of Mg lost as scrap [85,86]. For instance, extensive mechanical processing is necessary for producing components for the aerospace and automobile industries (oil pans, engine blocks, seat frames, transmission housings, etc.) [87]. Magnesium construction alloys for cyclically loaded structural applications include the AZ91D, NZK, AM-SC, and GW103 grades [88]. The elemental compositions of the ML10 and ML5 casting alloys generally correspond to those of the NZK (Mg–Nd–Zn–Zr) and AZ91D (Mg–Al–Zn) grades, respectively. ML5 is used in plane chassis, wings, control elements, and large-shaped castings such as helicopter gearbox housings (340 kg), compressor housings (720 kg), fan housings (290 kg), and other complex contoured parts. ML10 is implemented for the loaded components of instruments, engines, and equipment wherein leak tightness should be ensured [89]. Although in many countries aluminum and magnesium scrap are not classified as hazardous waste materials, their improper handling may result in accidental cases of fire and explosion [90,91]. Therefore, their effective utilization should be provided.

Summarizing the abovementioned, magnesium waste chips, shavings, and cuts have good potential for hydrogen generation. In a preceding comparative study [66], waste material samples were activated by ball milling (4 h) with 20 wt.% additives (KCl, Wood’s alloy, and their combination) and without additives and reacted with 3.5 wt.% NaCl aqueous solution. Unexpectedly, the highest specific hydrogen yields corresponded to the sample without additives, although it had the largest particles. That finding gave the idea that the effect of ball milling on the Mg scrap microstructural transformations and hydrogen generation performance should be studied in more detail, and that some other commercial alloys can be tested as additives promoting structural transformations and hydrogen evolution. 

The objective of the present study was to investigate the effects of ball milling duration and modification with a hard and brittle Devarda’s alloy (known for its easy crushing into powder) [92] on the Mg scrap microstructural transformations and hydrogen generation performance. Aqueous solutions with NaCl concentration of 3.5 wt.%, simulating sea water, were used as an oxidizing media. The present research was aimed to the optimization of ball milling time and additive amount to provide the effective transformation of Mg scrap into hydroreactive samples, ensuring high hydrogen yield and evolution rate. The results of the present study can be applied for Mg waste utilization with hydrogen production. 

## 2. Materials and Methods

The starting materials for hydroreactive samples were waste chips of the ML5 alloy with shavings of the ML10 grade (National State Standard GOST 2856-79) obtained from mechanical processing at an aircraft manufacturing plant, and reagent-grade Devarda’s alloy composed of 45 wt.% Al, 50 wt.% Cu, and 5 wt.% Zn (Technical Specification No. 6-09-36, ‘Rushim’ LLC, Moscow, Russia). Salt aqueous solution was prepared using deionized water and chemically pure NaCl salt (National State Standard GOST 4233-77, ‘VEKTON’ JSC, Saint-Petersburg, Russia).

Prior to ball milling, scrap particles were degreased with pure acetonitrile (Technical Specification No. 2636-092-44493179-04, ‘EKOS-1’ JSC, Moscow, Russia). The process included two cycles of ultrasonic cleaning (1 h) in an ultrasonic bath sonicator (PSB-2835-05; ‘PSB-Gals’ Ltd., Moscow, Russia), and further stirring (1 h) of the ‘scrap–acetonitrile suspension’ via a magnetic mixer (C-MAG; ‘HS 7 IKA-Werke’ GmbH & Co. KG, Staufen, Germany). Between the cycles, the ‘used’ acetonitrile portion was changed for a fresh one. The degreased metal scrap was separated and dried at ambient temperature for 24 h.

Ball milling was performed using a centrifugal ball mill (S 100; ‘Retsch’ GmbH, Haan, Germany), 24 stainless steel milling balls (10 mm), and a 50 mL milling pot, which was filled in a glove box (G-BOX-F-290; ‘FUMATECH’ Ltd., Novosibirsk, Russia) under pure argon (99.993%, National State Standard GOST 10157-79, ‘NII KM’ Ltd., Moscow, Russia). The ball-to-powder mass ratio was 24:1, rotational speed was 580 rpm, and milling durations were 0.5, 1, 2, and 4 h.

The experimental procedure included pouring 400 mL of 3.5 wt.% NaCl solution into a reactor (500 mL, Simax glass) and its heating or cooling with a heater (CC-308B; ‘ONE Peter Huber Kältemaschinenbau’ GmbH, Offenburg, Germany) or cryothermostat (LOIP FT-311-80; ‘Laboratory Equipment and Instruments’ Ltd., Saint-Petersburg, Russia), under stirring with a magnetic mixer (C-MAG HS 7; ‘IKA-Werke’ GmbH & Co. KG, Staufen, Germany). Then, a sample (0.25 g) was loaded into the reactor. Resulting hydrogen passed through a Drexel flask into a glass vessel with water. Water, ejected by incoming hydrogen, was collected in a flask placed onto scales (ATL-8200d1-I; ‘Acculab Sartorius Group’, New York, USA), for which readings were continuously transmitted to a computer. The temperatures in the reactor and glass vessel with water and hydrogen were measured, respectively, with an L-type thermocouple (TP.KhK(L)-K11; ‘Relsib’ LLC, Novosibirsk, Russia) and a Pt100-type resistance temperature detector (TS-1288 F/11; ‘Elemer’ LLC, Podolsk, Russia) connected to a multichannel thermometer (TM 5103; ‘Elemer’ LLC, Podolsk, Russia). The atmospheric pressure was measured by a barometer (BTKSN-18; Technical Specification No. 1-099-20-85, ‘UTYOS’ JSC, Ulyanovsk, Russia). The solid reaction product was separated using a Bunsen flask, Buchner funnel, paper filter, and circulating water vacuum pump (SHZ-D III; ‘FAITHFUL Instrument Co.’, Ltd., Hebei, China), and dried at ambient temperature for 24 h.

The ejected water volume, temperature in the glass vessel, and atmospheric pressure were used to calculate the hydrogen volume values at standard conditions (Standard DIN 1343: 101,325 Pa, 0 °C), in accordance with the ideal gas law. The hydrogen yields represented the ratio of the calculated volumes to the theoretical maximum hydrogen volume determined from three experiments with degreased scrap and 5 wt.% HCl aqueous solution. For each powder sample and each temperature point, a series of three experiments was carried out. 

For the hydroreactive samples and solid reaction products, X-ray diffraction (XRD) analysis was performed using a ‘Difraey 401’ diffractometer (‘Scientific Instruments’ JSC, Saint Petersburg, Russia) with Cr-Kα radiation (0.22909 nm). The XRD patterns were processed using a database (Powder Diffraction File™) from the International Centre for Diffraction Data (ICDD). Visual investigation and particle size measurements were carried out by means of an optical microscope (Bio 6) equipped with a high-resolution camera (UCMOS 10000KPA; ‘Altami’ LLC, Saint Petersburg, Russia). Particle size measurements were carried out by means of ‘Altami Studio 3.5’ software and included ‘capturing’ particles in the image under brightfield illumination, their contouring, and calculation of the contour sizes (maximum Feret diameters) using the calibration data. The photographs of the particle clusters were captured using darkfield illumination. A NOVA NanoSem 650 scanning electron microscope (FEI Co., Hillsboro, OR, USA) with an annular backscattered electron detector was used to investigate the surface morphology by scanning electron microscopy (SEM) and to evaluate the composition by energy-dispersive X-ray spectroscopy (EDX), and the specific surface areas were determined using a surface area and pore size analyzer Nova 1200e (‘Quantachrome Instruments’ LLC, Boynton Beach, FL, USA). The data from low-temperature nitrogen sorption measurements were processed by means of ‘Quantachrome NovaWin’ software applying the Brunauer–Emmett–Teller (BET) equation.

A schematic diagram including the main operations with the original materials and samples is given in Figure 1.

## 3. Results and Discussion

### 3.1. Characterization of Hydroreactive Samples

#### 3.1.1. Particle Size Distributions

Figure 2 illustrates the photographs of the original materials and ball milled samples. Devarda’s alloy particles represented small crystalline particles with sharp edges. Mg scrap consisted of metal particles with the traces of mechanical processing (scratches, grooves, cracks, ragged edges), widely diverging in their sizes. Both sorts of samples, ball milled as they were and those modified with Devarda’s alloy, demonstrated similar transformations depending on activation time. The samples ball milled for 0.5 and 1 h contained a lot of large, flattened particles, while those activated for 2 and 4 h were composed of much smaller pieces, looking like compacted objects with round edges. The corresponding particle size distributions (in form of histograms and cumulative curves) are represented in Figure A1 (see Appendix A). The photographs and particle size distributions show that during mechanical activation, size reduction took place.

The data on the particle sizes, averaged over their numbers, for the original materials and all samples are listed in Table 1. After 0.5 h of ball milling, large-sized original scrap chips (~1400 μm) reduced their sizes to 277 and 188 μm correspondingly for the sample without additives and that with Devarda’s alloy. Then, they continued getting smaller, and 0.5 h later (1 h in total), achieved 155 and 89 μm respectively. Ball milling during 2 h resulted in further decrease in the particle sizes: 68 for the sample without additives, and 83 μm for that modified with Devarda’s alloy. Additionally, 2 h later (4 h in total), the particle size of the sample without additives was 36 μm, and for the samples containing 5 and 10 wt.% Devarda’s alloy, the sizes were 40 and 32 μm respectively.

After 0.5 and 1 h mechanical activation, the particle size values for the samples with Devarda’s alloy were substantially smaller than those for the samples without additives: 188 vs. 277 μm and 89 vs. 155 μm, respectively. Taking into account the irregularity in the cumulative curve shape for the 2 h activated sample with Devarda’s alloy (convolution in the range ~15–120 μm, see Figure A1), it can be suspected that a more accurate size evaluation over a greater number of particles could give a value less than 83 μm. Thus, the actual average particle size of that sample was probably close to or even less than the 68 μm of the sample without additives, with both of them falling between ~60–70 μm, and the particle sizes of all three samples after 4 h activation turned out to be quite close to each other: they fell between 30 and 40 μm. Based on these results, the following conclusion can be made. Perhaps, at the earlier ball milling stages (up to 1–2 h), easily crushable Devarda’s alloy promoted particle size reduction. It probably accelerated the entire particle structure transformation as well. However, after 2–4 h of activation, the said effects of Devarda’s alloy became minor. 

#### 3.1.2. Specific Surface Area Measurements

The specific surface areas, measured by the low-temperature nitrogen sorption method, are tabulated in Table 2. According to the results, the original Mg scrap, composed mainly of plane chips and cuts highly diverging in their sizes, had a surprisingly large specific surface area of 2.567 m^2^/g. For all the samples, the effect of ball milling was the reduction of their specific surface areas as compared to the original material. For ball milling durations of 0.5 and 1 h, the values for Mg scrap with Devarda’s alloy and without it were nearly 0.94, 1.08, 2.02 and 2.04 m^2^/g correspondingly. For the samples mechanically activated for 2 h, the results fell beyond the detection limit of 0.1 m^2^/g, and for the said compositions ball milled for 4 h, the specific surface areas raised to ~0.94 and 1.43 m^2^/g, respectively. As to the sample with 10 wt.% Devarda’s alloy, its ball milling for 4 h resulted in a ~0.38 m^2^/g specific surface area. Such non-monotonic dependencies of the specific surface areas on ball milling time may indicate some radical transformations in the particles’ structure.

In study [93], the investigated Mg-based powders steadily increased their particle sizes after 1, 2, and 4 h of ball milling (6 mm steel balls, 20:1 ball to powder mass ratio). However, in the said study, the original material was composed of spherical particles which were flattened by the steel ball impacts, as expected, thus enlarging their surface area. This was also seen in study [42], wherein spherical Mg particles were used as starting materials as well. For the activation, 5.1 mm steel balls with a 10:1 ball to powder ratio were used. The difference was that the average particle sizes increased up to 2 h of mechanical activation, due to their transformation into flattened ‘flakes’, and gradually decreased after further 1 and 2 h (3 and 4 h in total), during which they were rearranged into irregular-shaped solids. According to the results from [43], the specific surface area of the Mg-based powder, ball milled for 1 h as it was, and with 5 wt.% Ni and Co, increased from 0.26 to 0.724, 0.888, and 0.917 m^2^/g respectively. Ball milling conditions were the same: 5.1 mm steel balls with a 10:1 ball to powder ratio, and the results, reported in [44], demonstrated a gradual decrease in the specific surface area of Mg powder from ~0.7 m^2^/g (original and 0.5 h ball milled) to ~0.5 and ~0.1 m^2^/g after 3 and 10 h of activation, respectively. The said powders were prepared using two 11.1 mm and one 14.3 mm steel balls, with an 8:1 ball to powder ratio. The data given above prove that the evolution of disperse metal materials during ball milling may proceed in different ways. It may depend on the additives, size, and particle shape of the original materials, as well as the efficiency of impacts during ball milling, and is affected by the size and quantity of milling balls as well as the ‘ball-to-powder’ ratio [94].

#### 3.1.3. Investigation by Scanning Electron Microscopy

The SEM images of the hydroreactive samples are given in Figure 3. The morphology for all samples was investigated in a secondary electron (SE) mode, and the microstructure for some samples, modified with Devarda’s alloy, was examined in a backscattered electron (BSE) mode. 

In the BSE images, Figure 3e,i, corresponding respectively to the samples containing 5 and 10 wt.% Devarda’s alloy, the base phase (Mg) is shown in grey, and the modifier’s phase (mainly Al_2_Cu) is represented by white spots. As it can be seen, the modifier was distributed over the particles’ surfaces rather well.

According to the SE images (5000× and 30,000× magnifications), the samples generally had the same texture, apparently formed by tearing away small metal pieces and their merging with large particles by cold welding from steel balls’ impacts. The exceptions were the samples, with Devarda’s alloy and without it, that were ball milled for 1 h. About a quarter of all their particles had a specific texture. Figure 3b,f clearly illustrate crystalline structures on the surface of the said particles that may give evidence of some recrystallization processes.

The SE images with 500× magnification illustrate the transformation of the particles’ shape and structure. For both material sorts, with Devarda’s alloy and without it, the particles of the samples ball milled for 0.5 and 1 h were flat-shaped, and those of the samples ball milled for 2 and 4 h looked like complex solid shapes, formed by agglomerations of smaller pieces cold-welded to each other. In a preceding study [80], the evolution of aluminum powder particles’ structure during ball milling was investigated. It was established that in the beginning of the ball milling process, aluminum particles (initially representing solid shapes) were flattened and cold-welded by impacts of steel balls to form bigger plane-shaped pieces. After 1 h, due to the accumulated plastic deformations, they became brittle, which caused their fracturing into smaller particles. The highest hydrogen yields (pure water, 80 °C) were obtained for the samples ball milled for 4 and 7 h. They consisted of ‘laminated’ structures, formed by locally cold-welded, thin particles with free spaces between them. After 11 h of ball milling, the said structures were compressed, which led to a decrease in the specific surface area of the resulting ‘compacted’ particles.

As to the present results, and taking into account the specific surface area data listed above, the following assumption about the microstructural transformation can be made. At ball milling durations up to 1 h, originally thin particles of ductile magnesium alloy did not undergo any considerable flattening which could enlarge their surface areas. They were cold-welded to each other and to small metal pieces, and torn away by high-energy impacts of heavy steel balls, thus forming flake-like particles. In the interval from 1 to 2 h, embrittlement under numerous severe plastic deformations caused particles to break into smaller pieces. Due to cold welding, the said smaller pieces agglomerated into large solid shapes which then were compacted. The samples mechanically activated for 2 h possibly did not represent the abovementioned ‘lamellar’ structures, and that was the reason for a drastic reduction in the specific surface areas of those materials. Additionally, from 2 to 4 h, large particles were turning into smaller ones, which mostly looked more like a monolith rather than an agglomeration. Presumably, in that time interval, larger ‘agglomerates’ were exposed to further compacting by impacts from various directions, which made them fragile and caused their cracking into smaller ‘monoliths’. Additionally, due to their smaller sizes, the finally obtained ‘monoliths’ provided larger specific surface areas, as compared to the ‘agglomerates’. 

#### 3.1.4. XRD and EDX Analyses

The XRD patterns for the original materials and samples, ball milled as they were or together with Devarda’s alloy, are shown in Figure 4. The basic phase of the scrap and all samples was Mg–Al alloy, representing a solid solution of aluminum in the primary hexagonal close-packed magnesium lattice. The face-centered cubic Al structure was identified as well. As it has been mentioned above, the ML5 grade alloy is generally similar to the AZ91D alloy, which is composed mainly of α-phase Mg and β-phase Mg_17_Al_12_ [95]. According to the preceding paper [66], the identification of the Al phase was associated with the enrichment of the Mg_17_Al_12_ phase with aluminum after a thermal treatment (homogenization at 420 °C for 12 h and aging at 200 °C for 8 h), standard for the ML5 alloy. In report [96], the content of Mg_17_Al_12_ in an ML5 alloy sample was 5.31 wt.%, with the content of Al in the Mg_17_Al_12_ phase achieving as much as 3.42 wt.% of the total mass. The mentioned figures mean that the said phase contained ~64 wt.% Al instead of ~44 wt.% (according to its chemical composition) [97]. In addition, a good dissolution of Zn in the Mg_17_Al_12_ phase, with partial substitution of Al with Zn (e.g., ternary phase Φ-Mg_21_(Al, Zn)_17_), has been established [96,98]. In the present study, one more version was suggested. Mg–Al–Zn alloys are known to undergo galvanic corrosion due to the presence of the β-phase which is more ‘noble’ than the basic α-phase (if the Mg_17_Al_12_ phase is not too abundant to produce a ‘shielding’ effect on Mg grains). For that reason, a common method to protect such a type of alloy against corrosion is to form coatings of Al or Al-based compounds upon them [99,100]. So, as no detailed history of the scrap was provided, the detected Al phase might originate either from coating, or from enrichment with aluminum. As to the ML10 grade, no typical components (Nd, Zr) of that alloy were identified by the XRD analysis. However, their presence was confirmed by the EDX measurement results: Zr—0.53 wt.%, Nd—2.85 wt.% (Mg, Zn, C, and O—91.44, 0.50, 4.04, and 0.65 wt.% respectively). 

For Devarda’s alloy sample, an intermetallic θ-phase Al_2_Cu (tetragonal lattice), hexagonal Zn structure, and solid solution of Zn in Al (face-centered, cubic) were identified. The samples manufactured of scrap and Devarda’s alloy contained an Al_2_Cu phase. No Fe from steel balls was registered. However, the maximum Fe contamination accessed in the previous study [66] was nearly 0.1 wt.% (4 h ball milling). For several metal scrap peaks of the Mg–Al and Al phases, their intensities appeared to be higher than those for the samples ball milled for 2 and 4 h. Such a result can be explained by the partially textured structure (preferred crystal orientation along a certain crystallographic direction) of the original scrap. A closer inspection of the XRD patterns revealed that for the ball milled samples, the most notable basic phase peaks were slightly shifted rightward, towards larger angles, as compared with those of the original scrap. For the samples activated for 4 and 2 h, the full width at half maximum (FWHM) values of their basic phase peaks were the largest and second-largest. They were larger than those of the samples ball milled for 0.5 and 1 h, and most of the peaks for 1 h ball milling were wider than those for 0.5 h. The said peak broadening implies a decrease in crystalline size and a greater accumulation of microstrains in the lattice [93,101,102], and for the samples ball milled for 1 and 2 h, a new Al–Mg phase (Al_0.95_Mg_0.05_) was detected. This new phase started its formation after 0.5–1 h of ball milling and disappeared in the samples activated for 4 h. The micrographs in Figure 3f give evidence of the recrystallization process, induced apparently due to Mg dissolution in the detected Al phase under ball milling. The new phase contribution increased with ball milling time (from 1 to 2 h) for the samples of scrap only, and decreased for those modified with Devarda’s alloy.

### 3.2. Characterization of Hydroreactive Samples

#### 3.2.1. Effect of Devarda’s Alloy Content

The kinetic curves for the scrap samples (ball milled for 4 h with 5 and 10 wt.% Devarda’s alloy, and with no additives), tested under 15, 25 and 50 °C, are represented in Figure 5. As it can be seen, all curves have an S-shape typical for topochemical reactions. The said heterogeneous reactions are commonly characterized by an acceleration stage at the beginning (corresponding to the origination and growth of nucleation sites), stage of the highest reaction rate with further deceleration, and final termination stage.

The results for 15 °C demonstrated that the sample with no additives provided a hydrogen yield and maximum hydrogen evolution rate substantially lower than those for the samples with 5 and 10 wt.% Devarda’s alloy. The respective values were (58.27 ± 1.51)% vs. (74.04 ± 1.37) and (73.92 ± 1.47)%, and 16.8 mL/g/min vs. 49.6 and 67.5 mL/g/min. The hydrogen yields for the samples with the additive were almost the same after 4 h of measurements. At the beginning, the sample with the higher additive content reacted faster. 

At 25 °C, the difference between the hydrogen yields for all samples considerably decreased. The values of ‘scrap-to-hydrogen conversion’ for the samples with no additives, with 5, and 10 wt.% Devarda’s alloy, were correspondingly (75.82 ± 7.94), (90.97 ± 5.79), and (87.00 ± 5.63)%. The respective maximum reaction rates increased to 35.5, 99.0, and 134.6 mL/g/min.

After 2 h experiments under 50 °C, the sample with no additives, and the samples containing 5, and 10 wt.% Devarda’s alloy, provided hydrogen yields of (79.91 ± 6.19), (93.87 ± 0.22), and (94.98 ± 3.18)% correspondingly. The respective fastest hydrogen production rates were 74.9, 301.0, and 414.0 mL/g/min. 

For the investigated samples, increasing the reaction temperature from 25 to 50 °C did not provide a large increase in the final hydrogen yield values, but considerably accelerated their reaction processes. Modification of the scrap with Devarda’s alloy enhanced both hydrogen yields and reaction rates. The higher additive content (10 wt.%) provided somewhat better results. However, 5 wt.% seemed to be enough for improving the scrap’s hydrogen generation performance. 

Earlier studies [41,42,43,44,93] proved that in conductive media (NaCl or KCl solution), Mg-based materials, ball milled with some metals (e.g., Ni, Co, Al, Fe, Bi), had their less-noble Mg component attacked by galvanic corrosion. The effect of Devarda’s alloy was similar, and consisted in the formation of micro-galvanic cells between a nobler Al_2_Cu phase (nobler than Mg and Al [93,103]) and the basic Mg phase. Some amount of Al was likely affected by the anodic dissolution in the NaCl aqueous solution as well. 

#### 3.2.2. Effect of Ball Milling Duration

The dependency of oxidation kinetics on ball milling duration for the samples, of scrap only, and with 5 wt.% Devarda’s alloy is illustrated in Figure 6. The data are given for three tested temperatures. The kinetic curves have a typical S-shape as well. For most of them, the initial acceleration section was reduced so much that it seemed indistinguishable under the represented time scale. 

For the samples without additives tested at 15 °C, the lowest hydrogen yield, (58.27 ± 1.51)%, and maximum reaction rate, 16.8 mL/g/min, corresponded to 4 h of ball milling. The second-lowest values were obtained for 0.5 h activation: (80.75 ± 1.91)% and 27.0 mL/g/min. The samples, ball milled for 1 and 2 h had the highest hydrogen yields of (84.51 ± 2.60)% and (86.11 ± 2.89)%, close to each other. Their relevant maximum reaction rates achieved 41.8 and 32.1 mL/g/min. As to the sample modified with Devarda’s alloy, the lowest results of (74.04 ± 1.37)% ‘scrap-to-hydrogen conversion’ and 49.6 mL/g/min generation rate corresponded to the 4 h activated sample as well. The results for other ball milling durations 0.5, 1, and 2 h were substantially close to each other. The respective hydrogen yields were (91.97 ± 2.19), (93.48 ± 1.21), and (90.64 ± 1.83)%, while the fastest reaction rates increased to 73.1, 87.3, and 67.4 mL/g/min.

The experimental results obtained under 25 °C demonstrated decreased divergences between the curves for both samples: with scrap only, and with Devarda’s alloy. The samples without additives, ball milled for 0.5 and 4 h, had their error bars overlapped. Their corresponding hydrogen yield and maximum reaction rate values were lower than those for 1 and 2 h samples, and achieved (83.17 ± 0.92)%, 43.5 mL/g/min and (75.82 ± 7.94)%, 35.5 mL/g/min. Activation for 1 and 2 h provided the highest and second-highest yields of (89.77 ± 1.50)% and (86.70 ± 1.53)%, with the rates of 63.0 and 40.0 mL/g/min. After 1 h of experiment, the hydrogen yield values obtained for the samples ball milled for 1, 2, and 4 h were slightly lower than those obtained in study [93] for Mg powder. The divergences can be explained by different compositions of the scrap and pure Mg. Modification with Devarda’s alloy again provided the increased hydrogen yield and reaction rate values. Although at the beginning the 4 h activated sample ‘lagged behind’ in hydrogen generation, the eventual hydrogen yields for that milling time, 2, 1, and 0.5 h were correspondingly (90.97 ± 5.79), (91.23 ± 1.75), (91.40 ± 0.60), and (91.74 ± 1.58)%, with the fastest rates of 99.0, 129.6, 158.5, and 147.5 mL/g/min. 

The kinetic curves for the samples, of scrap only, obtained at 50 °C, demonstrated generally the same relative positioning. Although this time the curve for 4 h topped the 0.5 h curve, they matched within their error bars. Their relevant hydrogen yields were (79.91 ± 6.19) and (79.95 ± 1.36)%, with the fastest reaction rates of 74.9 and 75.4 mL/g/min. The ‘scrap-to-hydrogen conversion degrees’ for 1 and 2 h ball milling durations again were the highest and second-highest: (91.12 ± 1.19) and (91.03 ± 1.41)%. The respective maximum reaction rates achieved 143.4 and 73.5 mL/g/min, and for the samples with the additive, the eventual amounts of released hydrogen were close to each other and achieved as much as (92.71 ± 1.27), (95.36 ± 0.38), (94.00 ± 2.09), and (93.87 ± 0.22)% for 0.5, 1, 2, and 4 h of ball milling, respectively, and the relevant fastest hydrogen evolution rates were as high as 412.2, 470.9, 342.7, and 301.0 mL/g/min.

As it can be concluded from the summarization of the above data, addition of Devarda’s alloy increased hydrogen yields and production rates for all tested ball milling durations and temperatures. For all sample groups with the same composition, reacting under the same temperature, the samples mechanically activated for 4 h provided majorly the lowest hydrogen yields and slowest reaction progress. Such an effect might be caused by a partial oxidation of the powder particles with residual oxygen during a prolonged ball milling. Thus, the formation of MgO during ball milling was established in study [44]. For their Mg powder, ball milled for 1 h, the measured MgO content was 1 wt.% only, but after another 2 and 7 h of ball milling (3 and 10 h in total), that value increased to 10 and 12 wt.%. In that study, the original material represented powder with the average particle size of nearly 44 μm, while in the present study, much larger scrap particles were used. Therefore, their oxidation was apparently much less intensive. However, the oxidation of the particles’ surface could be the reason for their slow reaction rates. Taking into account the temperature-dependent lagging, it might be assumed that at lower temperatures, the lag was larger due to a slower hydration of MgO, while at higher temperatures it became smaller, due to faster MgO hydration with the formation of a less dense Mg(OH)_2_ [104]. As to the relatively ‘humble’ results for 0.5 h activated samples without additives, they might be ascribed to the lower strain energy accumulated during ball milling as compared to the samples activated for a longer time, combined with the absence of the ‘cathode’ Al_2_Cu phase-enhancing galvanic corrosion. The highest performance, corresponding to 1 h ball milled samples, might be explained by their large specific surface areas together with the accumulated crystal lattice structure imperfections, and the second-highest values for the samples, activated for 2 h, were possibly associated with the combination of the greater strain energy accumulation and less severe oxidation with residual oxygen during ball milling.

### 3.3. Characterization of Solid Reaction Products

#### 3.3.1. Investigation by Scanning Electron Microscopy

The SEM images of the reaction products for the samples ball milled for 1 and 4 h with Devarda’s alloy and without additives are shown in Figure 7. The microphotographs under 10,000× magnification illustrate the surface textures of the samples. The said textures were generally formed by clusters of large-sized ‘flakes’ and spongy regions, composed of much smaller structure elements looking like moss cover. Within the captured areas, 1 h ball milled samples had greater number of the mentioned clusters formed by large-sized ‘flakes’ than those activated for 4 h. Although mere visual assessment of the sizes of large-scale ‘flakes’ was not sufficient for making any conclusions, some observations were worth to be communicated. The reaction products captured under higher magnifications were compared. The ‘flakes’ of the reacted sample ball milled for 1 h without additives were the largest, while those with Devarda’s alloy seemed to be the second-largest. It was found as well that the largest ‘flakes’, corresponding to 1 h ball milled samples, were generally larger than those for 4 h activation. The surface structures, constituted by larger ‘flakes’, were presumably more liquid-permeable than those formed by smaller ‘flakes’ or representing ‘mossy’ regions.

#### 3.3.2. X-ray Diffraction Analysis

The XRD patterns for the solid reaction products from the oxidation of 1 h ball milled hydroreactive samples are illustrated in Figure 8. The results demonstrated that both reaction products contained NaCl (contamination from the aqueous solution) and an unreacted Al–Mg phase (solid solution of Mg in Al). The major component was Mg(OH)_2_ (hexagonal structure), resulting from the oxidation of Mg (metal scrap basic component) in the sodium chloride aqueous solution. From the comparison between the XRD patterns for the original samples and their reaction products, it can be seen that the Mg–Al and Al phases disappeared. The absence of the said components was likely attributed either to their being exhausted by the reaction, or to their low contents (beyond the detection limits). No reaction products for Al were detected as well. Such a result might be associated with a relatively low Al content in the scrap, or with the probable formation of pseudoboehmite characterized by a weak diffraction intensity.

### 3.4. Summarization and Discussion of the Results

The data on hydrogen yields and maximum reaction rates, for all experiments, are given in Figure 9 and tabulated in Table 3 (for references). In accordance with the presented results, almost for all ball milling durations and tested temperatures, hydrogen yields for the samples with Devarda’s alloy were higher than those for the samples of scrap only. The exceptions were the following combinations of conditions: 1 h and 25 °C, 2 h and 15 °C, 4 h and 25 °C. The least hydrogen yields majorly corresponded to the samples mechanically activated for 4 h. The exceptions were the samples with 5 wt.% Devarda’s alloy tested under 25 and 50 °C. The uncertainties in the eventual hydrogen yield values might be attributed to the non-uniform particle size distributions of the samples, inhomogeneous allocation of additive pieces, structural imperfections among particles, and various amounts of residual oxygen in the milling pot. 

The fastest hydrogen generation rates rose to their maximum values, as ball milling time increased from 0.5 to 1 h. After 2 h of ball milling, they decreased to the values close to those for 0.5 h (samples of scrap only), or lower (samples with additive), and dropped to their minimum values after another 2 h of activation (4 h in total). Again, the highest values for the samples, modified with Devarda’s alloy, topped those for the samples of scrap only. 

From the summarization of all the information given above, the following key findings can be derived. Implementation of Devarda’s alloy improved the hydrogen generation performance of the metal scrap, due to the formation of ‘microgalvanic cells’ between the nobler additive and basic phases. They favored a severe corrosion of Mg (and partially Al) in the conductive media (NaCl solution). The modified samples provided higher hydrogen yields and reaction rates within the whole range of tested temperatures and ball milling durations. Ball milling duration also affected the hydroreactive properties of the samples. Thus, the highest results for all tested samples, of scrap only and with Devarda’s alloy, were obtained for 1 h mechanical activation. The main effect of ball milling constituted the structural transformations of the particles. Mechanical activation resulted in the reduction in the particle sizes (which, during the first 2 h, was faster for the samples with Devarda’s alloy), and non-monotonic change in the specific surface areas. The highest specific surface area corresponded to the original scrap. After ball milling for 0.5 and 1 h, it decreased by approximately 20% for the samples of scrap only, and more than halved for the samples with Devarda’s alloy. The specific surface areas for the samples, activated for 2 h, were beyond the detection limits. However, after another 2 h of activation (4 h in total), they raised again: to the value twice lower than that for 0.5 and 1 h for the samples without additives, and to a value nearly 50% higher as compared to those for 0.5 and 1 h for the modified samples. The most important visible structural transformations were particles’ evolution from the large flat-shaped ‘flakes’ (0.5–1 h milling) to solid-shaped compacted objects (2–4 h milling), for which sizes considerably reduced during the latest 2 h of activation. The broadening of the basic phase peaks for the scrap-based samples, activated for 2 and 4 h, over those for the 0.5 and 1 h ball milled samples pointed to their greater accumulation of strain energy.

In study [66], 4 h ball milled samples with different compositions, based on the same scrap, were tested in the same solution. According to the microphotographs, all the samples were composed of solid-shaped compacted particles. The smallest ones (6 μm in average) corresponded to the sample ball milled with KCl, and the largest (36 μm) were from the sample with no additives. Although no specific surface area results were reported, it might be expected that the smaller particles had a higher specific surface area than the larger ones. Therefore, the finer hydroreactive powder was expected to provide a higher hydrogen yield, but it provided a higher reaction rate only. Even the cavities, originating after salt particles’ dissolution, and contributing to the specific surface area of the modified sample, did not significantly improve its ‘modest’ result. By a closer investigation of the XRD data, it was found that the sample ball milled with no additives had basic phase peaks notably broader than those of the powders modified with KCl. It was assumed that the greater number of accumulated microstrains resulted in a better sustaining of the reaction with the corrosive solution over time, while a larger surface area accelerated the reaction progress in the beginning. 

It can be assumed that the highest hydrogen yields and evolution rates for the 1 h ball milled samples resulted from the combination of the following factors. On the one hand, their specific surface areas exceeded those of the samples activated for longer time, and on the other hand, a greater accumulation of lattice imperfections, as compared with the samples activated for shorter time, enhanced corrosion in the tested NaCl solution. The samples ball milled for 2 h had the lowest specific surface areas, but accumulated a lot of lattice defects during ball milling. Although the 4 h ball milled samples had a larger specific surface area and higher accumulated strain energy than those activated for 2 h, their hydrogen production performance decreased, probably because of their partial oxidation with residual oxygen during a prolonged ball milling. Additionally, despite the 0.5 and 1 h ball milled samples modified with Devarda’s alloy having lower specific surface areas, as compared to those of scrap only, the contribution of the nobler ‘cathode’ phase Al_2_Cu to the entire hydrogen generation performance presumably occurred higher than that of the highly extended surface.

## 4. Conclusions

A number of hydroreactive samples were manufactured of Mg scrap with no additives, with 5, and 10 wt.% Devarda’s alloy, by ball milling for 0.5, 1, 2, and 4 h to study the effects of ball milling duration and additive content on their structural evolution and hydrogen generation properties in 3.5 wt.% NaCl solution.

For both sorts of samples, modified with Devarda’s alloy and of scrap only, the major effects of ball milling constituted in the accumulation of crystal lattice imperfections, oxidation of Mg with residual oxygen, and mirostructural evolution of scrap particles. The accumulation of crystal lattice imperfections favored pitting corrosion of Mg in the conductive NaCl solution. The oxidation of Mg with residual oxygen during prolonged ball milling resulted in a significant reduction in hydrogen evolution rates for the samples ball milled for 4 h. The microstructural evolution consisted in the radical transformation in the particles’ shapes and sizes: from large plane-shaped pieces (original scrap, 0.5 and 1 h ball milling) into small, compacted, solid-shaped objects (2 and 4 h ball milling). It was notable that the specific surface area of the samples ball milled for 0.5 and 1 h reduced as compared with that of the original scrap; it dropped beyond the detection limits for 2 h ball milled samples and it increased for the 4 h activated samples. The microstructural evolution stages, apparently, included cold welding of the plane-shaped particles during the first 0.5–1 h of ball milling; their fracturing and compacting into solid-shaped ‘agglomerates’ after 1–2 h; crushing of the ‘agglomerates’ into smaller ‘monolithic’ pieces by the end of 4 h of activation. 

The effects of Devarda’s alloy additive constituted in the formation of ‘microgalvanic cells’ between the base Mg and nobler Al_2_Cu phases and acceleration of the abovementioned microstructural evolution at the early ball milling stages (0.5–1 h). The ‘microgalvanic cells’ enhanced galvanic corrosion of Mg in the conductive salt solution. Due to the hard and brittle nature of Devarda’s alloy, its particles likely assisted the ball milling of the scrap and provided faster reduction in the scrap particles’ size. Although the addition of 10 wt.% Devarda’s alloy provided the highest initial reaction rates, the gain of 5 wt.% additive was satisfying as well.

The maximum hydrogen yields and evolution rates for both sample sorts, modified with Devarda’s alloy and with no additives, corresponded to the milling time of 1 h, while the lowest values were obtained for 4 h of activation. The highest results for 1 h activated samples were caused by the combination of the highest specific surface area and greater accumulation of microstrains, favoring corrosion, during ball milling (and ‘microgalvanic cells’ for the sample with the additive), and the lowest performance of the samples activated for 4 h was attributed to their potential oxidation with residual oxygen during a prolonged ball milling. 

Thus, the optimal milling time (1 h) and Devarda’s alloy content (5 wt.%) for hydroreactive samples were established. The results of the present study can be used for Mg waste utilization with hydrogen generation.

## Figures and Tables

**Figure 1 materials-15-08058-f001:**
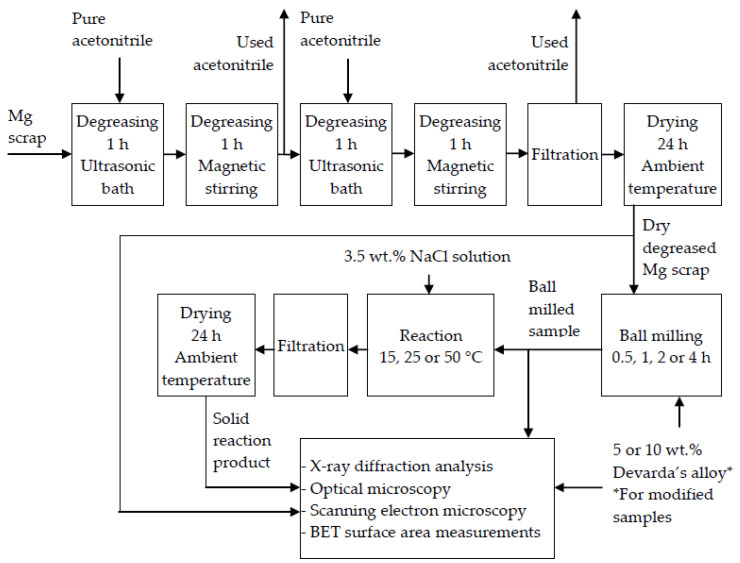
Schematic diagram of the operations with the original materials and samples.

**Figure 2 materials-15-08058-f002:**
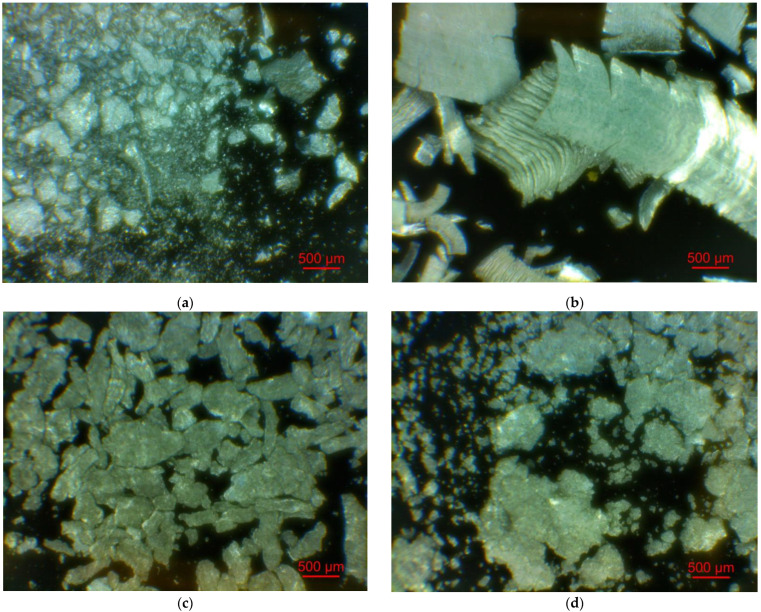
Optical microscope images of the original scrap, Devarda’s alloy and hydroreactive samples with different compositions and ball milling durations: (**a**) Devarda’s alloy; (**b**) Mg scrap; (**c**) Mg scrap, 0.5 h; (**d**) Mg scrap, 1 h; (**e**) Mg scrap, 2 h; (**f**) Mg scrap, 4 h; (**g**) Mg scrap + 5 wt.% Devarda’s alloy, 0.5 h; (**h**) Mg scrap + 5 wt.% Devarda’s alloy, 1 h; (**i**) Mg scrap + 5 wt.% Devarda’s alloy, 2 h; (**j**) Mg scrap + 5 wt.% Devarda’s alloy, 4 h; (**k**) Mg scrap + 10 wt.% Devarda’s alloy, 4 h.

**Figure 3 materials-15-08058-f003:**
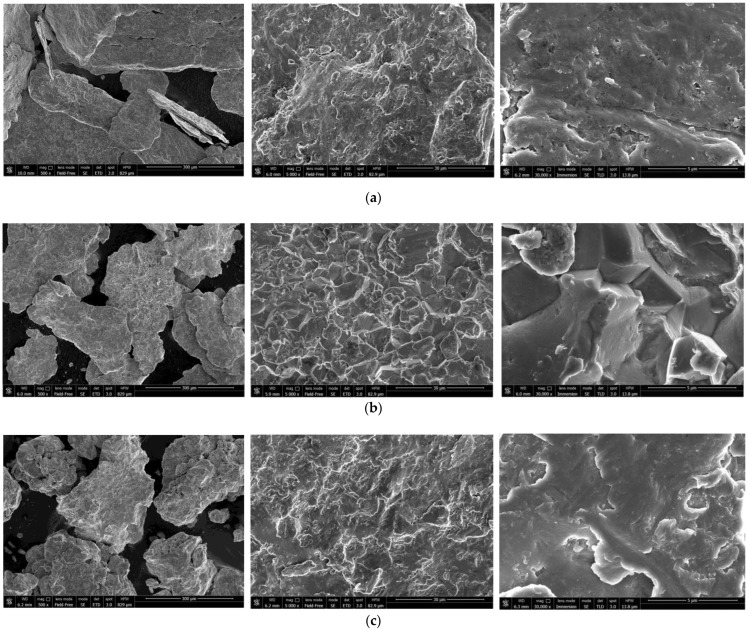
SEM images of the hydroreactive samples with different compositions and ball milling durations: (**a**) Mg scrap, 0.5 h; (**b**) Mg scrap, 1 h; (**c**) Mg scrap, 2 h; (**d**) Mg scrap, 4 h; (**e**) Mg scrap + 5 wt.% Devarda’s alloy, 0.5 h; (**f**) Mg scrap + 5 wt.% Devarda’s alloy, 1 h; (**g**) Mg scrap + 5 wt.% Devarda’s alloy, 2 h; (**h**) Mg scrap + 5 wt.% Devarda’s alloy, 4 h; (**i**) Mg scrap + 10 wt.% Devarda’s alloy, 4 h.

**Figure 4 materials-15-08058-f004:**
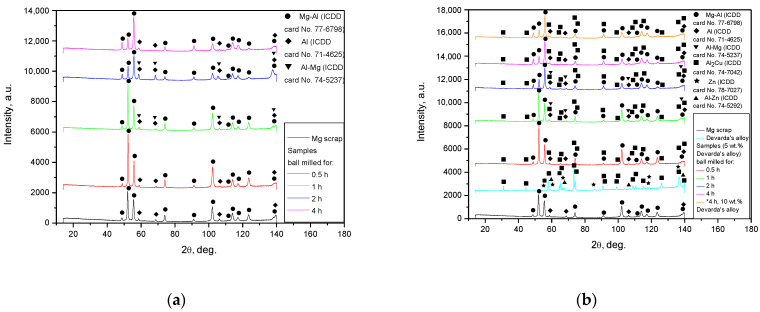
XRD patterns of the original materials and ball milled samples: (**a**) Mg scrap and samples without additives; (**b**) Mg scrap, Devarda’s alloy and samples composed of them.

**Figure 5 materials-15-08058-f005:**
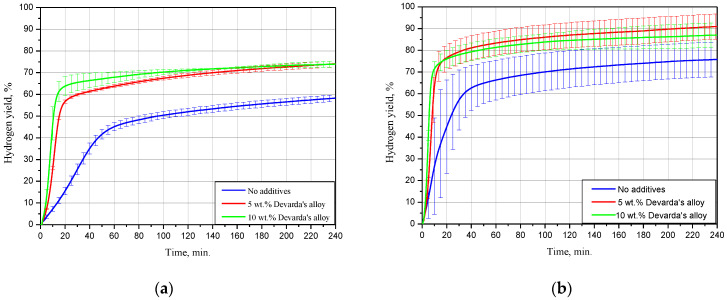
Hydrogen evolution kinetic curves for the samples with various compositions under different temperatures: (**a**) 15 °C; (**b**) 25 °C; (**c**) 50 °C.

**Figure 6 materials-15-08058-f006:**
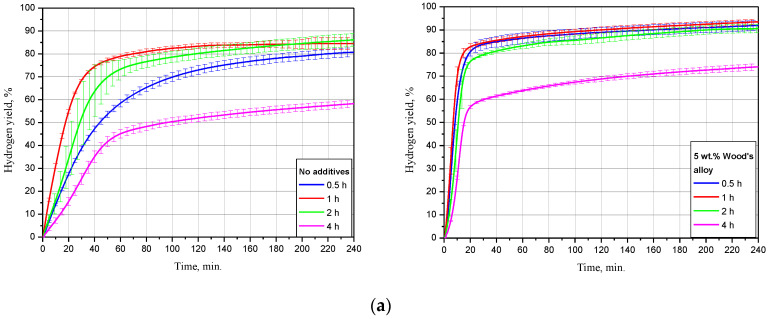
Hydrogen evolution kinetic curves for the samples with various ball milling durations under different temperatures: (**a**) 15 °C; (**b**) 25 °C; (**c**) 50 °C.

**Figure 7 materials-15-08058-f007:**
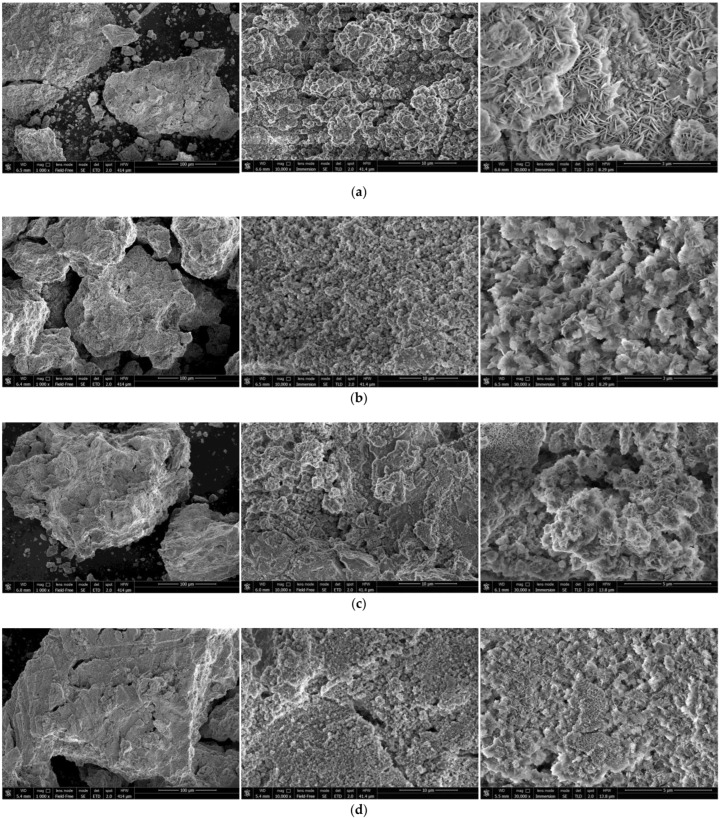
SEM images of the reaction products originated from different hydroreactive samples: (**a**) 1 h ball milled, no additives; (**b**) 4 h ball milled, no additives; (**c**) 1 h ball milled, 5 wt.% Devarda’s alloy; (**d**) 4 h ball milled, 5 wt.% Devarda’s alloy.

**Figure 8 materials-15-08058-f008:**
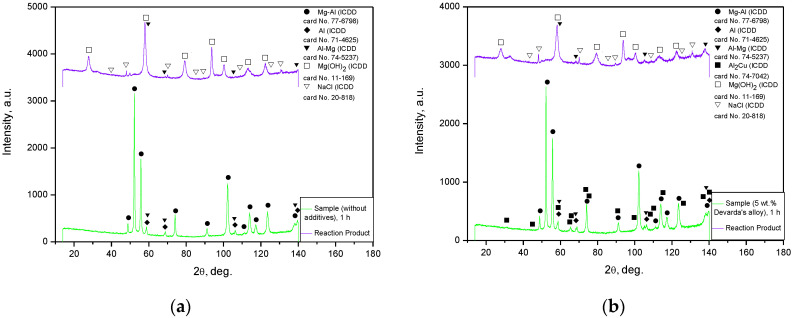
XRD patterns of the samples and corresponding reaction products: (**a**) sample ball milled 1 h without additives and its reaction product; (**b**) sample ball milled 1 h with 5 wt.% Devarda’s alloy and its reaction product.

**Figure 9 materials-15-08058-f009:**
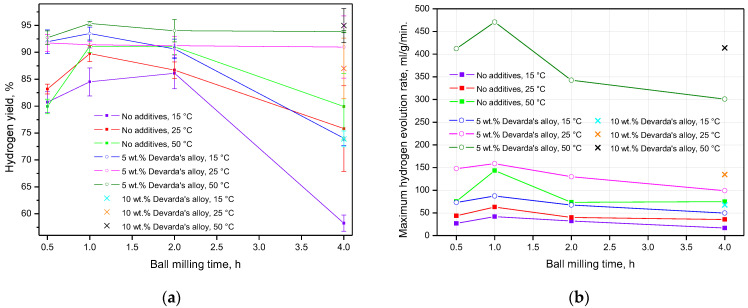
Summarized experimental data for the samples with various compositions and different reaction temperatures vs. ball milling time: (**a**) hydrogen yields; (**b**) maximum hydrogen evolution rates.

**Table 1 materials-15-08058-t001:** Average particle sizes of the original materials and samples with different ball milling durations and compositions.

Material	Average Particle Size, μm
Original Mg scrap	~1400
Devarda’s alloy	36
Mg scrap (0.5 h)	277
Mg scrap (1 h)	155
Mg scrap (2 h)	68
Mg scrap (4 h)	36
Mg scrap + 5 wt.% Devarda’s alloy (0.5 h)	188
Mg scrap + 5 wt.% Devarda’s alloy (1 h)	89
Mg scrap + 5 wt.% Devarda’s alloy (2 h)	83
Mg scrap + 5 wt.% Devarda’s alloy (4 h)	40
Mg scrap + 10 wt.% Devarda’s alloy (4 h)	32

**Table 2 materials-15-08058-t002:** Specific surface areas of the original Mg scrap and samples with different ball milling durations and compositions.

Material	Specific Surface Area, m^2^/g
Original Mg scrap	2.567
Mg scrap (0.5 h)	2.036
Mg scrap (1 h)	2.022
Mg scrap (2 h)	<0.1
Mg scrap (4 h)	0.943
Mg scrap + 5 wt.% Devarda’s alloy (0.5 h)	0.944
Mg scrap + 5 wt.% Devarda’s alloy (1 h)	1.076
Mg scrap + 5 wt.% Devarda’s alloy (2 h)	<0.1
Mg scrap + 5 wt.% Devarda’s alloy (4 h)	1.460
Mg scrap + 10 wt.% Devarda’s alloy (4 h)	0.375

**Table 3 materials-15-08058-t003:** Summarized experimental data for the samples with various compositions, ball milling durations and different reaction temperatures.

Composition	Ball Milling Time, h	Temperature, °C	Hydrogen Yield, %	Maximum Reaction Rate, mL/g/min
No additives	0.5	15	80.75 ± 1.91	27.0
		25	83.17 ± 0.92	43.5
		50	79.95 ± 1.36	75.4
	1	15	84.51 ± 2.60	41.8
		25	89.77 ± 1.50	63.0
		50	91.12 ± 1.19	143.4
	2	15	86.11 ± 2.89	32.1
		25	86.70 ± 1.53	40.0
		50	91.03 ± 1.41	73.5
	4	15	58.27 ± 1.51	16.8
		25	75.82 ± 7.94	35.5
		50	79.91 ± 6.19	74.9
Scrap + 5 wt.% Devarda’s alloy	0.5	15	91.97 ± 2.19	73.1
		25	91.74 ± 1.58	147.5
		50	92.71 ± 1.27	412.2
	1	15	93.48 ± 1.21	87.3
		25	91.40 ± 0.60	158.5
		50	95.36 ± 0.38	470.9
	2	15	90.64 ± 1.83	67.4
		25	91.23 ± 1.75	129.6
		50	94.00 ± 2.09	342.7
	4	15	74.04 ± 1.37	49.6
		25	90.97 ± 5.79	99.0
		50	93.87 ± 0.22	301.0
Scrap + 10 wt.% Devarda’s alloy	4	15	73.92 ± 1.47	67.5
		25	87.00 ± 5.63	134.6
		50	94.98 ± 3.18	414.0

## Data Availability

Not applicable.

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
