# Peer review of "Microstructural Transformation and Hydrogen Generation Performance of Magnesium Scrap Ball Milled with Devarda’s Alloy"

_materials, 2022, doi:10.3390/ma15228058_

Round 1

Reviewer 1 Report

In this work, the authors revealed the influence of ball milling duration and modification on sample structural transformation and hydrogen yield. This is an interesting field, but some issues need to be noted:

1.     In the title and Lines 93 and 94, the authors focused on the effect of ball milling duration and the sample structural transformation on the hydrogen performance. But the samples’ structural evolution also derived from the influence of ball milling conditions. Therefore, the title of this work may be “Effects of Magnesium Scrap Modification with Devarda’s Alloy during Ball Milling on Structural Transformations and the Hydrogen Generation Performance”.

2.     In the keywords, the authors selected “microstructure”. So the structural transformation could be “microstructural transformation”.

Author Response

Greetings, Dear Reviewer!

Thank you very much for your attention to our manuscript. Your comments were helpful in improving the title – one of the key points of each paper aimed to attract attention of the potential audience.

The list of corrections, improvements and updates provided in accordance with your comments is attached herein.

No.

Comment

Response

1

In the title and Lines 93 and 94, the authors focused on the effect of ball milling duration and the sample structural transformation on the hydrogen performance. But the samples’ structural evolution also derived from the influence of ball milling conditions. Therefore, the title of this work may be “Effects of Magnesium Scrap Modification with Devarda’s Alloy during Ball Milling on Structural Transformations and the Hydrogen Generation Performance”.

Many thanks for the advice. Indeed, ball milling conditions affected both microstructural transformation and hydrogen generation performance. In accordance with your suggestion and one from another reviewer (to make the title shorter for a better clarity for the readers), a new version was ‘elaborated’. The updated title is ‘Microstructural Transformation and Hydrogen Generation Performance of Magnesium Scrap Ball Milled with Devarda’s Alloy’. 

2

In the keywords, the authors selected “microstructure”. So the structural transformation could be “microstructural transformation”.

According to the suggestion, ‘structural transformation’ was turned into ‘microstructural transformation’.

Reviewer 2 Report

The present paper entitled “Effects of Magnesium Scrap Modification with Devarda’s Alloy and Structural Transformations during Ball Milling on its Hydrogen Generation Performance” is having potential information for publications. The present paper is related to current topic of energy utilization. The authors have carried out extensive experiments and analyses. I am recommending to accept the article with minor revision. The authors have to address the following points.

1.  In the abstract, the ratio or wt.% or vol.% of Al5 (Mg based alloy) and Devarda’s alloy is to be mentioned clearly. In the present one is not clear.

2. The main objectives of the present study are to be incorporated at the end of introduction

3. A schematic diagram representing the entire work may be incorporated in section 2 of materials and method

4. The scale marker mentioned in Figure 1 is not visible. Moreover, each image is taken at different magnification. It is recommended to use same scale so that it will be easy for comparison

5. Some precipitates / phases are observed in Figure 2 (middle image) which has to be marked in the corresponding images

Author Response

Greetings, Dear Reviewer!

We, authors, would like to thank you for your kind attention to our study. Your detailed inspection of the manuscript and resulting suggestions contributed a lot to its improvement.

The list of corrections, improvements and updates, provided in accordance with your comments, is attached herein.

No.

Comment

Response

1

In the abstract, the ratio or wt.% or vol.% of Al5 (Mg based alloy) and Devarda’s alloy is to be mentioned clearly. In the present one is not clear.

Thank you for your attention. The content values for the sample components (in wt.%) were added.

2

The main objectives of the present study are to be incorporated at the end of introduction

The objective and aim of the present study were incorporated at the end of Introduction.

3

A schematic diagram representing the entire work may be incorporated in section 2 of materials and method

The schematic diagram, including the main operations with the original materials and samples, was added in Section 2.

4

The scale marker mentioned in Figure 1 is not visible. Moreover, each image is taken at different magnification. It is recommended to use same scale so that it will be easy for comparison

The photographs of the powders and original materials were redesigned. The same scale and a more contrast color were used. For better arrangement, the particle size distribution histograms were moved to Appendix A.

5

Some precipitates / phases are observed in Figure 2 (middle image) which has to be marked in the corresponding images

The phases on the SEM images (BSE mode) were marked.

Reviewer 3 Report

The effects of magnesium scrap modification with Devarda’s alloy and structural transformations during ball milling on its hydrogen generation performance was shown in this work.

In the work the hydroreactive samples maked of Mg-based scrap by its ball milling. Microstrains (which facilitated pitting corrosion, and the formation of microgalvanic cells) were removed by ball milling.

This allowed to increase the efficiency of hydrogen production.

The results of the measurements proved the correctness of the undertaken research direction.

The maximum hydrogen yields and evolution rates for sample types corresponded to the milling time of 1 h. In work was noted that long ball milling is the probable reason for that was oxidation with residual oxygen in the milling pot during a prolonged ball milling. As a consequence, this oxidation lowers the yield of hydrogen production.

Some suggestion follows:

- it is necesarry to rewrite the Abstract chapter to emphasize the purpose of the study and to indicate for the reader the potential application of the solution already in Abstract

- it is necesarry to add and expand the information on the potential application of the solution in the Introduction chapter

- it is necesarry to rewrite the Conclusion chapter 

Author Response

Greetings, Dear Reviewer!

We, authors, are grateful for your attention to our study. Your useful comments helped us to improve the manuscript.

The list of corrections, improvements and updates, provided in accordance with your comments, is attached herein.

No.

Comment

Response

1

it is necesarry to rewrite the Abstract chapter to emphasize the purpose of the study and to indicate for the reader the potential application of the solution already in Abstract

The Abstract was rewritten. The purpose and potential application were indicated.

2

it is necesarry to add and expand the information on the potential application of the solution in the Introduction chapter

The information on the potential application of the solution was added to the Introduction chapter.

3

it is necesarry to rewrite the Conclusion chapter 

The conclusion chapter was rewritten. The key findings were highlighted.

Reviewer 4 Report

The work evaluates the effects of various technical parameters (such as ball milling time and structural alteration) on the hydroreactive samples and probes their hydrogen generation performance. The topic is interesting. The results are properly supported with sufficient discussions, but the quality of figures is mostly weak. Authors should redesign the figures and use larger fonts for better legibility for the future readers. The manuscript can be accepted after answering the following comments:

1- Title can be shorter for a better clarity for the readers.

2- The manuscript needs extensive English grammatical corrections.

3- Abstract needs careful reconsideration and rewriting. Please only highlight the principal concepts of this study and its critical findings.

4- Figures 1,2,6: The scale bars in the optical and SEM images are not clear. Please refer to the format of the SEM images in these articles (ACS Nano 2022, 16, 12606; Energy Storage Materials 47 (2022) 61; Energy Storage Materials 47 (2022) 51). If possible, please redesign the images to be presented in a single page for better clarity. It’s unusual to trace a single figure in multiple pages. Moreover, what do the authors mean by 100 MKM (scale bars in Figure 1)? Scale bars should be changed to “X00 microns” or “X00 mm”.

5- Figure 1: particle size distribution histograms can be moved to the supplement or presented as an insets along with optical images in the main manuscript. In either cases, the fonts must be extremely larger for greater legibility.

Author Response

Greetings, Dear Reviewer!

Thank you very much for your keen interest in our study. Your accurate examination of the manuscript and resulting helpful comments contributed a lot to its improvement.

The list of corrections, improvements and updates, provided in accordance with your comments, is attached herein.

No.

Comment

Response

1

Title can be shorter for a better clarity for the readers.

In accordance with your advice, a new version was ‘elaborated’. The updated title is ‘Microstructural Transformation and Hydrogen Generation Performance of Magnesium Scrap Ball Milled with Devarda’s Alloy’. 

2

The manuscript needs extensive English grammatical corrections.

The manuscript text was edited; a number of grammatical corrections were made.

However, the revised versions of all MDPI manuscripts are usually checked by a journal’s specialist, native English speaker. Therefore, most likely, the final version for publication will differ from the current revised version.

3

Abstract needs careful reconsideration and rewriting. Please only highlight the principal concepts of this study and its critical findings.

The abstract was reconsidered and rewritten. The principal concepts of the study and its critical findings were highlighted, and the less important information was excluded.

4

Figures 1,2,6: The scale bars in the optical and SEM images are not clear. Please refer to the format of the SEM images in these articles (ACS Nano 2022, 16, 12606; Energy Storage Materials 47 (2022) 61; Energy Storage Materials 47 (2022) 51). If possible, please redesign the images to be presented in a single page for better clarity. It’s unusual to trace a single figure in multiple pages. Moreover, what do the authors mean by 100 MKM (scale bars in Figure 1)? Scale bars should be changed to “X00 microns” or “X00 mm”.

The photographs of the powders and original materials (Figure 1) were redesigned. The same scale and a more contrast color were used. For better arrangement, the particle size distribution histograms were moved to Appendix A. MKM was changed to μm by changing the language settings.

As to the SEM images, due to the technical maintenance of our local equipment, their creation was outsourced. Unfortunately, we have not tools to edit them. The images were enlarged for better legibility of the scale bars. Due to the use of journal’s format templates, it is hardly possible to shift the images closer to one another. As all of the represented images are important, we cannot reduce their number, because those from the first column show particle’s shape, the second column contains photographs taken under BSE mode, and the third column of images clearly illustrates recrystallization.

However, the final arrangement of all the images is usually performed by the journal specialists. In the web pages of published MDPI papers, all sets of figures (and tables) can be selected for browsing under proper magnification. Thus, convenient access to all the graphical data will be provided.

5

Figure 1: particle size distribution histograms can be moved to the supplement or presented as an insets along with optical images in the main manuscript. In either cases, the fonts must be extremely larger for greater legibility.

The particle size distribution histograms were moved to Appendix A. Larger font was employed.

Round 2

Reviewer 3 Report

The manuscript has been improved sufficiently.

I believe that the paper needs no further corrections. Thus, I recommend this work for publication.

Reviewer 4 Report

I checked the authors' corrections and modifications. The manuscript can be accepted for publication in Materials in the current format from my viewpoint.